# Site-Selective Solvation-Induced Conformational Switching of Heteroleptic Heteronuclear Tb(III) and Y(III) Trisphthalocyaninates for the Control of Their Magnetic Anisotropy

**DOI:** 10.3390/molecules28114474

**Published:** 2023-05-31

**Authors:** Alexander G. Martynov, Kirill P. Birin, Gayane A. Kirakosyan, Yulia G. Gorbunova, Aslan Yu. Tsivadze

**Affiliations:** 1Frumkin Institute of Physical Chemistry and Electrochemistry, Russian Academy of Sciences, Leninsky pr., 31, Building 4, 119071 Moscow, Russia; kirill.birin@gmail.com (K.P.B.); gayakira@mail.ru (G.A.K.); yulia@igic.ras.ru (Y.G.G.); tsiv@phyche.ac.ru (A.Y.T.); 2Kurnakov Institute of General and Inorganic Chemistry, Russian Academy of Sciences, Leninsky pr., 31, 119991 Moscow, Russia

**Keywords:** triple-decker phthalocyaninate, terbium, lanthanide-induced shift, axial magnetic anisotropy, conformational switching, solvation

## Abstract

In the present work, we report the synthesis of isomeric heteronuclear terbium(III) and yttrium(III) triple-decker phthalocyaninates **[(BuO)_8_Pc]M[(BuO)_8_Pc]M*[(15C5)_4_Pc]** (M = Tb, M* = Y or M = Y, M* = Tb, [(BuO)_8_Pc]^2−^–octa-*n*-butoxyphthalocyaninato-ligand, [(15C5)_4_Pc]^2−^–tetra-15-crown-5-phthalocyaninato-ligand). We show that these complexes undergo solvation-induced switching: the conformers in which both metal centers are in square-antiprismatic environments are stabilized in toluene, whereas in dichloromethane, the metal centers M and M* are in distorted prismatic and antiprismatic environments, respectively. This conclusion follows from the detailed analysis of lanthanide-induced shifts in ^1^H NMR spectra, which makes it possible to extract the axial component of the magnetic susceptibility tensor $\chi_{\mathrm{ax}}^{\mathrm{Tb}}$ and to show that this term is particularly sensitive to conformational switching when terbium(III) ion is placed in the switchable “M” site. This result provides a new tool for controlling the magnetic properties of lanthanide complexes with phthalocyanine ligands.

## 1. Introduction

Within the wide variety of coordination compounds containing paramagnetic metal centers, lanthanide complexes occupy a special place because of their unique optical and magnetic properties, which can be fine-tuned by changing the ligand environment [1,2,3,4,5,6,7,8]. Understanding the correlations between the composition and symmetry of the coordination sphere of lanthanide ions is the ultimate prerequisite for providing such tuning on a rational basis [9,10].

One of the manifestations of the magnetic properties of lanthanide ions is the shift of resonance signals in the NMR spectra of their complexes in comparison with the spectra of isostructural diamagnetic counterparts [11]. The sign and magnitude of such a lanthanide-induced shift (LIS, ${\Delta \delta }_{k}$) of a resonating nucleus “k” depend on the nature of both the lanthanide ion and the ligand, and can be separated into the isotropic through-bond (contact, ${\Delta \delta }_{k}^{c}$ [12,13]) and anisotropic through-space (dipolar or pseudo-contact, ${\Delta \delta }_{k}^{pc}$ [14,15]) contributions.
(1)$${\Delta \delta }_{k}=\delta_{k}^{para}{-\delta }_{k}^{dia}={\Delta \delta }_{k}^{c}{+\Delta \delta }_{k}^{pc}$$

Importantly, the dipolar component is typically dominant in the total LIS value; moreover, it is functionally related to the geometry of the complex [16,17]. This anisotropic part of LIS results from the removal of the spherical symmetry of the lanthanide ions Ln^3+^ upon the formation of coordination compounds. Thus, being placed into the origin, these ions form the principal magnetic axis system where the internal polar coordinates $\theta_{k}$, $\varphi_{k}$ and $r_{k}$ of the resonating nucleus can be considered. In the case of the axially symmetric complexes possessing at least a three-fold symmetry axis, the functional correlation between the dipolar LIS, the magnetic properties of the lanthanide ion and the geometry of the complex can be expressed as:(2)$${\Delta \delta }_{k}^{pc}=\frac{\chi_{ax}^{Ln}}{12\pi }\cdot G_{k}$$

The geometric parameter $G_{k}$ is a function of $r_{k}$—the distance between the lanthanide ion and the resonating nucleus k, and $\theta_{k}$—an angle between the vector $\left(\overset{-}{{Ln}^{3+},k}\right)$ and the main symmetry axis:(3)$$G_{k}=\frac{3{cos}^{2}\theta_{k}-1}{r_{k}^{3}}$$

The powerful Equation (2) suggests that NMR spectroscopy of paramagnetic lanthanide complexes is not limited to routine confirmation of the composition and purity of newly synthesized compounds, but can also be used to study the geometric structure of complexes in solution [18,19]. This feature is useful, for instance, in structural biology by applying lanthanide probes introduced into biomolecules [20,21,22]. Furthermore, NMR can provide information on the magnetic properties of lanthanide ions in a given coordination environment through the term $\chi_{ax}^{Ln}$, which is an axial component of the magnetic susceptibility tensor. This information is complementary to the data typically obtained from magnetochemical studies [23,24,25,26], and in this context, NMR spectroscopy can be used as a more affordable analytical tool to guide the selection of complexes for further advanced measurements.

In the present work, we used ^1^H NMR spectroscopy to study the magnetic properties of the new heteronuclear trisphthalocyaninates **[(BuO)_8_Pc]M[(BuO)_8_Pc]M*[(15C5)_4_Pc]**, where M ≠ M* are Tb or Y, [(BuO)_8_Pc]^2-^ and [(15C5)_4_Pc]^2-^ are octa-*n*-butoxy- or tetra-15-crown-5-phthalocyaninato ligands. For brevity, these ligands are henceforth designated as [B_4_] and [C_4_], where the letters “B” and “C” stand for BuO- and 15C5-substituted phthalic units in the phthalocyanine rings, respectively, so that in this notation, the target complexes will be designated as **[B_4_]M[B_4_]M*[C_4_]**.

The interest to characterize the magnetic properties of these complexes using ^1^H NMR arises from their specific conformational behavior depending on the solvation environment. Thus, for the examples of the homonuclear complexes with M = M* = Tb or Y, we have previously shown that the fragments [B_4_]M[B_4_] can adopt either staggered or gauche conformations in aromatic or halogenated aliphatic solvents, respectively [27]. In turn, it switches the coordination polyhedron of the metal center M from square-antiprismatic (SAP) to distorted prismatic (DP). In contrast, the fragment [B_4_]M*[C_4_] is conformationally invariant—it exists in the staggered conformation in both types of solvents, so the metal center M* is always in the SAP environment. The difference in the conformational states of these complexes results from solvent–solvate interactions stabilizing either a staggered or a gauche arrangement of the adjacent ligand. It was definitively explained using single crystal X-ray diffraction experiments performed for the solvates **[B_4_]Y[B_4_]Y[C_4_]·10CH_2_Cl_2_** or **[B_4_]Y[B_4_]Y[C_4_]·13C_7_H_8_** (Figure 1) [27]. Spectroscopic signatures of both gauche and staggered conformers in solutions were identified using UV-vis and DFT calculations on the examples of homoleptic complexes **M_2_[B_4_]_3_** and **M_2_[C_4_]_3_** (M = Tb, Y [28]) together with ^1^H NMR performed for Eu(III) counterparts [29].

Thus, in the present work, the availability of structural data providing the geometric parameters $G_{k}$ for two conformers of **[B_4_]Y[B_4_]Y[C_4_]** allowed us to extract the axial anisotropy terms $\chi_{ax}^{Tb}$ from the ^1^H NMR spectra of the heteronuclear Tb(III)-containing complexes in different solvents and to show that their magnetic anisotropy can be tuned by controlling their conformational state.

Apart from the interest in correlations between the structure and magnetic properties of lanthanide complexes, the tuning of anisotropy provides some useful practical outcomes. For example, we have previously demonstrated the thermosensing properties of a wide range of paramagnetic complexes of lanthanides with tetra-15-crown-5-phthalocyanine **Ln_2_[C_4_]_3_**, Ln = Nd, Tb, Dy, Ho, Er, Tm for in situ NMR thermometry [30,31,32]. It was shown that the best sensitivity gain up to 1.1 ppm/K was obtained for the Tb(III) complex, which shows the highest anisotropy.

## 2. Results

The synthesis of the target heteronuclear complexes **[B_4_]M[B_4_]M*[C_4_]**, where M ≠ M* are Tb or Y was straightforward (Figure 1) due to the previously reported procedure for the homonuclear counterparts [27]. Briefly, butoxy-substituted double-deckers **M[B_4_]_2_** (M = Tb or Y) were treated with tetra-15-crown-5-phthalocyanine **H_2_[C_4_]** and acetylacetonates bearing another metal ion M*(acac)_3_·*n*H_2_O (M* = Y or Tb) in the refluxing mixture of 1,2,4-trichlorobenzene and 1-octanol (9:1 v/v). The resulting target complexes were readily separated in high yields using column chromatography on alumina from the unreacted starting double-deckers and the sole by-products—homonuclear trisphthalocyaninates **M*_2_[C_4_]_3_**.

The isolated isomeric complexes were characterized using a variety of physicochemical methods. MALDI-TOF MS confirmed the presence of the desired set of phthalocyanine ligands and metal ions (Appendix A), but apparently failed to distinguish between the isomers.

UV-vis characterization of the complexes was performed in toluene, as a representative of aromatic solvents, and dichloromethane as a halogenated alkane (Figure 2 and Appendix A). Thus, in toluene, both complexes showed well-resolved intense split Q-bands at 643 and 696 nm together with less intense Soret and N-bands at 362 and 293 nm. In contrast, the spectra in CH_2_Cl_2_ were dramatically different—the intensity of their strongly broadened Q-bands was significantly decreased in comparison with the Soret and N-bands. Several inflexions were observed on both the long- and short-wavelength sides of the Q-bands. Overall, the observed solvatochromic behavior was consistent with the existence of the synthesized complexes in different conformers in the studied solvents, namely, fully staggered in toluene (Figure 1a) and gauche/staggered in both CH_2_Cl_2_ and CHCl_3_ (Figure 1b) [27].

The synthesized heteronuclear complexes were characterized using ^1^H NMR in deuterated toluene and dichloromethane. Due to the paramagnetic nature of the Tb^3+^ ions, the resonance signals in the spectra of these complexes were spread over wide ranges of chemical shifts—from strongly positive to very strongly negative, and these ranges also depended on the solvent used for recording the spectra (Figure 3).

To assign these spectra, we used the transformation of equation (2), which suggests that if we consider LIS to be essentially dipolar, then the ratio of LIS for a pair of protons, $H_{k}$ and $H_{l}$, can be approximated using a ratio of their geometric parameters.
(4)$$\frac{{\Delta \delta }_{k}}{{\Delta \delta }_{l}}\approx \frac{{\Delta \delta }_{k}^{pc}}{{\Delta \delta }_{l}^{pc}}=\frac{G_{k}}{G_{l}}\equiv R_{kl}$$

In turn, Equation (4) suggests that the approximate position of the resonance signals of the protons $H_{k}$ can be calculated from the resonance signal of at least one firmly assigned proton $H_{l}$ in the spectrum of the paramagnetic complex:(5)$$\delta_{k}^{para}\approx \delta_{k}^{dia}+{\Delta \delta }_{l}\cdot R_{kl}$$

The geometric parameters $G_{k,l}$ were obtained by averaging the polar coordinates of the selected protons in the structures of solvates of **[B_4_]Y[B_4_]Y[C_4_]** with either toluene or dichloromethane; thus, providing the axially symmetric structures that can be considered as models of the heteronuclear complexes in solutions [33]. The set of diamagnetic chemical shifts was obtained from the spectra of **[B_4_]Y[B_4_]Y[C_4_]** measured in the corresponding solvents [27].

The aromatic protons of the phthalocyanine macrocycles and the methylene protons of the substituents proximal to the Pc ligands were used for further analysis. In all cases, the largest absolute values of $G_{k}$ corresponded to the aromatic protons of the inner phthalocyanine ligand bH_Pc_^i^, so that the most upfield shifted signal was assigned to these protons. In turn, it allowed us to assign the rest of the required signals. The accuracy of the assignments was checked using ^1^H-^1^H COSY (Appendix A), and in general, the plots of the calculated chemical shifts vs. the experimental values were characterized by perfect linearities with R^2^ greater than 0.99. Altogether, these results justified the validity of the dipolar approximation of LIS for the heteronuclear complexes studied herein.

Plotting the averaged coordinates of the selected protons on the contour maps of G(r;θ) (Figure 4) gives a clear graphical explanation as to why some signals in the spectra of heteronuclear complexes are shifted upfield and most of them are shifted downfield (entries in bold and regular font styles in Table 1). This is because protons get into areas with either negative or positive values of the function G(r;θ) [33,34].

Finally, the availability of structural and NMR data for both conformers of two heteronuclear complexes allowed us to find the axial component of the magnetic susceptibility tensors $\chi_{ax}^{Tb}$ to correlate it with the symmetry of the coordination polyhedron of the Tb^3+^ ions. With this aim, the dependencies of LIS vs. $G_{k}$ were plotted and least square linearization was used to find the slopes of these dependencies and convert them into $\chi_{ax}^{Tb}$ in accordance with equation 2 (Figure 5a,b).

The derived values of $\chi_{ax}^{Tb}$ (Figure 5c) clearly show that switching between two conformers has a profound effect on the magnetic properties of the Tb^3+^ ions, and the magnitude of this effect depends on whether it is placed in the switchable site [B_4_]/[B_4_] or the invariant site [B_4_]/[C_4_]. Thus, the transition from toluene-*d*_8_ to CD_2_Cl_2_ in the case of **[B_4_]Tb[B_4_]Y[C_4_]** due to the switching of the Tb^3+^ coordination polyhedron from SAP to DP increases $\chi_{ax}^{Tb}$ by 23%—from 7.77 ± 0.18 × 10^−31^ to 9.56 ± 0.26 × 10^−31^ m^3^.

Interestingly, although the polyhedron of the paramagnetic center in **[B_4_]Y[B_4_]Tb[C_4_]** is not switched, minor structural perturbations of its coordination sphere associated with the overall reorganization of the molecule also cause a smaller but still noticeable increase in axial anisotropy $\chi_{ax}^{Tb}$ by 10%—from 8.20 ± 0.28 × 10^−31^ to 9.02 ± 0.20 × 10^−31^ m^3^. These results suggest that the effects of symmetry breaking and coordination sphere reorganization act simultaneously [35] and further in-depth analysis using quantum-chemical calculations may shed light on the contribution of each of these effects to the control of the axial anisotropy.

## 3. Discussion

Previously, we have demonstrated the possibility of tuning the axial anisotropy of Tb^3+^ ions introduced into heteroleptic crown-substituted trisphthalocyaninates–**[C_4_]M*[C_4_]M(Pc)**, where M and M* = Tb or Y [36]. These complexes have been shown to act as supramolecular receptors with switchable rotational states—in the native state, both metal centers M and M* adopt square antiprismatic environments with a twist angle of 45° between the adjacent macrocycles. However, the addition of potassium cations resulted in their intercalation between the crown-substituted decks, reducing the twist angle to zero and providing the M* center with a square prismatic (SP) environment. Similar to the results studied here, the change from SAP to SP also caused a spectacular increase in the $\chi_{ax}^{Tb}$ by 25%—from 8.36 ± 0.15 × 10^−31^ to 10.63 ± 0.27 × 10^−31^ m^3^. In contrast, the square-antiprismatic polyhedron of the M metal center remained intact upon binding of K^+^ cations, and such binding has a much smaller effect on $\chi_{ax}$ of Tb^3+^ in this site—it increased from 9.43 ± 0.19 × 10^−31^ to 9.61 ± 0.16 × 10^−31^ m^3^.

The correlations between the magnetic behavior of single molecules and the magnitude of the anisotropy have also been emphasized by several authors [37,38,39,40]. For example, the triple-decker binuclear Tb(III) complex with fused phthalocyaninate ligands is characterized by record-high values of both the energy barrier for spin reversal, U_eff_ (588 cm^−1^), and the axial magnetic anisotropy $\chi_{ax}^{Tb}$ (10.39 × 10^−31^ m^3^), which is achieved by the geometric spin arrangement [37]. For comparison, a significantly lower value of U_eff_—230 cm^−1^ was found for diterbium(III) tris-octabutoxyphthalocyaninate **Tb_2_[B_4_]_3_**, which is also characterized by lower $\chi_{ax}^{Tb}$—0.86 × 10^−30^ m^3^ [41].

Taken together, these results suggest that control over the rotational state of phthalocyanine ligands in sandwich complexes together with their magnetic anisotropy can be used to control their magnetic properties and that these complexes are attractive models for studying the influence of both large and small molecular motions on the magnetic properties of lanthanide complexes. Thus, further magnetochemical measurements of the synthesized heteronuclear complexes will be useful to verify these correlations, paving the way to the rational design of magnetic materials via anisotropy tuning.

Finally, due to the presence of crown-ether substituents in the synthesized complexes, they can be used as molecular building blocks to form supramolecular dimers in the presence of alkali metal ions [42,43] to study the long-range interactions between paramagnetic metal centers.

## 4. Materials and Methods

### 4.1. Materials

Starting phthalocyanines **Y[B_4_]_2_**, **Tb[B_4_]_2_** and **H_2_[C_4_]** were synthesized according to the previously reported procedures [44,45]. 1,2,4-trichlorobenzene (TClB, for synthesis, Sigma-Aldrich), 1-octanol (for synthesis, Sigma-Aldrich, Burlington, MA, USA), yttrium(III) and terbium(III) acetylacetonanes (99.95 and 99.9%, respectively, Sigma-Aldrich), neutral alumina (50–200 μm, Macherey-Nagel, Düren, Germany) were used as received from the commercial suppliers. Chloroform (reagent grade, Ekos-1, Staraya Kupavna, Russia) was distilled over CaH_2_.

### 4.2. Methods

Matrix-assisted laser desorption ionization time-of-flight (MALDI-TOF) mass spectra were measured on a Bruker Daltonics Ultraflex spectrometer. Mass spectra were registered in positive ion mode using 2,5-dihydroxybenzoic acid as a matrix. UV-vis spectra in the 250–900 nm range were measured using a JASCO V-770 spectrophotometer in quartz cells with a 0.5–1 cm optical path. NMR spectra were recorded using a Bruker Avance III spectrometer with a 600 MHz proton frequency at 303 K with the residual solvent resonances as internal references (δ toluene 7.09 ppm, dichloromethane 5.32 ppm). Typically, 5 mg of complexes were dissolved in 0.6 mL of the corresponding deuterated solvent to provide ca. 2.3 mM concentration. The applied deuterated dichloromethane (99.5 atom% D, ABCR, Karlsruhe, Germany) and chloroform (99.8 atom% D, ZEOchem, Uetikon am See, Switzerland) were filtered prior to use through the Pasteur pipettes filled with alumina to remove possible acidic impurities. Deuterated toluene (99.5 atom% D, ABCR) was used without additional purification.

^1^H NMR spectra were acquired with a standard Bruker *zg30* pulse program for a 30-degree flip angle. The acquisition and relaxation delays were 1 s and the number of scans was 32. The spectra were recorded with 192,298 points resolution and a line broadening factor of −0.5 Hz for Fourier transformation. ^1^H-^1^H COSY spectra were acquired with a standard Bruker gradient-enhanced quantum-filtered COSY pulse sequence *cosygpqf*. The acquisition and relaxation delays were 0.137 s and 1 s in each scan, respectively, with 4 scans per increment. The spectra were recorded with 16,384 × 512 points resolution.

### 4.3. Synthesis and Characterization of the Triple-Decker Complexes

***Trisphthalocyaninate [B_4_]Y[B_4_]Tb[C_4_].*** Yttrium(III) bis(octa-butoxyphthalocyaninate) **Y[B_4_]_2_** (88 mg, 39 μmol) and tetra-15-crown-5-phthalocyanine **H_2_[C_4_]** (62 mg, 49 μmol) were dissolved in a mixture of 4.5 mL 1,2,4-trichlorobenzene and 0.5 mL 1-octanol. The resulting solution was brought to gentle reflux under a slow stream of argon and terbium acetylacetonate (69 mg, 0.15 mmol) was added. After 7 min, the consumption of the starting reagents stopped, as evidenced using UV-vis spectroscopy; the reaction mixture was cooled to room temperature and the resulting dark blue solution was transferred to the chromatographic column filled with alumina in a mixture of chloroform and hexane (1:1 *v*/*v*). Gradient elution with a mixture of CHCl_3_ with hexane followed by a mixture of CHCl_3_ with 0 → 2% methanol afforded the target complex as a dark blue powder (94 mg, yield 65%). MALDI TOF MS: *m/z* calculated for C_192_H_232_N_24_O_36_TbY 3699.5, found 3700.5 [M^+^]. UV-vis (Toluene) λ_max_ (nm) (log ε): 696 (4.75), 643 (5.53), 527 (4.36), 362 (5.34), 292 (5.21). UV-vis (CH_2_Cl_2_) λ_max_ (nm) (log ε): 641 (5.05), 553 (4.58), 352 (5.25), 293 (5.20). ^1^H NMR (600 MHz, Toluene-*d*_8_) δ 25.22 (s, 8H, bH_Pc_^o^), 9.47 (d, 8H, *J* = 58.5 Hz, 1^o^’), 1.48 (d, *J* = 58.7 Hz, 8H, 1^o^), 0.48—−0.05 (m, 32H, 2^o,o^’ and 3^o,o^’), −0.23 (s, 24H, CH_3_^o^), −2.07, −2.61, −3.02, −4.73 (4s, 4 × 8H, γ^o,o^’ and δ^o,o^’), −7.53 (s, 8H, β^o^’), −10.48 (s, 8H, β^o^), −10.75 (s, 24H, CH_3_^i^), −13.29 (d, *J* = 44.0 Hz, 8H, α^o^’), −14.47 and −14.78 (2s, 2 × 8H, 3^ib,ic^), −15.77 and −16.38 (2s, 2 × 8H, 2^ic,ic^), −23.23 (d, *J* = 68.0 Hz, 8H, 1^ib^), −25.23 (d, *J* = 52.0 Hz, 8H, 1^ic^), −36.46 (d, *J* = 66.6 Hz, 8H, α^o^), −52.05 (s, 8H, cH_Pc_^o^), −64.58 (s, 8H, bH_Pc_^i^). ^1^H NMR (600 MHz, Methylene Chloride-*d*_2_) δ 25.91 (s, 8H, bH_Pc_^o^), 9.05 (s, 8H, 1^o^’), 0.9—−0.24 (br m, 64H, 1^o^, 2^o,o^’, 3^o,o^’ and CH_3_^o^), −8.41 (br s, 24H, CH_3_^i^), −12.03 and −12.26 (2s, 2 × 8H, 3^ib,ci^), −14.15 and −14.79 (2s, 2 × 8H, 2^ib,ic^), −18.19 (br s, 8H, 1^ib^), −20.68 (br s, 8H, α^o^’), −30.95 (br s, 8H, α^o^), −32.98 (1^ic^), −67.00 (br s, 8H, cH_Pc_^o^), −68.21 (s, 8H, bH_Pc_^i^).

***Trisphthalocyaninate [B_4_]Y[B_4_]Tb[C_4_].*** Terbium(III) bis(octa-butoxyphthalocyaninate) **Tb[B_4_]_2_** (88 mg, 38 μmol) and tetra-15-crown-5-phthalocyanine **H_2_[C_4_]** (60 mg, 47 μmol) were dissolved in a mixture of 4.5 mL 1,2,4-trichlorobenzene and 0.5 mL 1-octanol. The resulting solution was brought to gentle reflux under a slow stream of argon and terbium acetylacetonate (57 mg, 0.14 mmol) was added. After 7 min, the consumption of the starting reagents stopped, as evidenced using UV-vis spectroscopy, the reaction mixture was cooled to room temperature and the resulting dark blue solution was transferred to the chromatographic column filled with alumina in a mixture of chloroform and hexane (1:1 *v*/*v*). Gradient elution with a mixture of CHCl_3_ with hexane followed by a mixture of CHCl_3_ with 0 → 2% methanol afforded the target complex as a dark blue powder (102 mg, yield 74%). MALDI TOF MS: *m/z* calculated for C_192_H_232_N_24_O_36_TbY 3699.5, found 3699.1 [M^+^]. UV-vis (Toluene) λ_max_ (nm) (log ε): 696 (4.74), 643 (5.53), 527 (4.34), 362 (5.33), 293 (5.20). UV-vis (CH_2_Cl_2_) λ_max_ (nm) (log ε): 644 (5.10), 545 (4.57), 352 (5.26), 293 (5.21). ^1^H NMR (600 MHz, Toluene-*d*_8_) δ 24.81 (s, 8H, cH_Pc_^i^), 9.85 (s, 8H, α^o^’), 4.02 and 1.61 (2s, 2 × 8H, β^o^’ and β^o^), 4.17, 3.73, 3.30, 2.45 (4s, 4 × 8H, γ^o,o^’ and δ^o,o^’), 2.09 (overlapped with CHD_2_ signal of toluene-*d*_8_, α^o^), −6.56 (s, 24H, CH_3_^o^), −9.44 (s, 16H, 3^o,o^’), −9.79 (s, 24H, CH_3_^i^), −11.28 and −11.17 (2s, 2 × 8H, 2^o,o^’), −13.13 and −13.55 (2s, 2 × 8H, 3^ib,ic^), −13.80 (d, *J* = 46.2 Hz, 8H, 1^o^’), −14.35 and −15.02 (2d, *J* = 25 Hz, 2 × 8H, 2^ib,ic^), −20.90 (d, *J* = 64.5 Hz, 8H, 1^ic^), −25.65 (d, *J* = 51.4 Hz, 8H, 1^o^), −33.64 (d, *J* = 63.4 Hz, 8H, 1^ib^), −50.96 (s, 8H, bH_Pc_^o^), −59.33 (d, 8H, bH_Pc_^i^). ^1^H NMR (600 MHz, Methylene Chloride-*d*_2_) δ 26.74 (s, 8H, cH_Pc_^i^), 10.69 (s, 8H, α^o^’), 3—−0.3 (br m, 56H, α^o^, β^o,o^’, γ^o,o^’ and δ^o,o^’), −8.82 (s, 24H, CH_3_^o^), −12.52 (d, *J* = 59.0 Hz, 16H, 3^o,o^’), −13.15 (s, 24H, CH_3_^i^), −14.81 and −15.41 (2d, *J* = 32.9 and 27.3 Hz, 2 × 8H, 2^o,o^’), −18.08 and −18.32 (2s, 2 × 8H, 3^ib,ic^), −20.27 and −20.63 (2s, 2 × 8H, 2^ib,ic^), −21.76 (d, *J* = 77.5 Hz, 8H, 1^o^’), −28.93 (d, *J* = 62.9 Hz, 8H, 1^ic^), −33.19 (d, *J* = 76.3 Hz, 8H, 1^o^), −43.80 (d, *J* = 83.1 Hz, 8H, 1^ib^), −69.66 (s, 8H, bH_Pc_^o^), −79.56 (s, 8H, bH_Pc_^i^).

## Data Availability

Data are contained within the article or Appendix A.

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
