# Peer review of "Site-Selective Solvation-Induced Conformational Switching of Heteroleptic Heteronuclear Tb(III) and Y(III) Trisphthalocyaninates for the Control of Their Magnetic Anisotropy"

_molecules, 2023, doi:10.3390/molecules28114474_

Round 1

Reviewer 1 Report

This manuscript "Site-selective solvation-induced conformational switching of heteroleptic heteronuclear Tb(III) and Y(III) trisphthalocyaninates for the control of their magnetic anisotropy" presents an interesting study on the correlation between the structure and magnetic properties of heteroleptic crown-substituted trisphthalocyaninates [C4]M*[C4]M(Pc). The authors demonstrate how changes in the rotational state of the phthalocyanine ligands can be used to tune the axial anisotropy of the Tb ions, resulting in an increase of 25%. This work is well-written and provides a thorough analysis of the results, as well as an insightful discussion on the implications of this work and potential applications in the design of magnetic materials. The authors have done a great job in this work, though there are a few minor revisions that could be made to improve the clarity of the manuscript. -For example, the "M and M*" should include the symbols for the elements, and the description of the native state of the complex could be more concise. -Additionally, it may be helpful to add a brief description of the NMR measurements to section 4.2. Methods (Pulse width, Number of scans, Spectral width, Relaxation Delay etc). -Please add information concerning study of lanthanide complexes in buffer solutions to introdaction [10.1021/acs.inorgchem.0c03082]. Overall, this is an interesting and well-written manuscript that provides an in-depth look at the correlation between the structure and magnetic properties of heteroleptic crown-substituted trisphthalocyaninates. With the suggested revision, this manuscript is ready for publication.

Scientific sound corrections (not essential, but will help make your manuscript clear to understand):

"in  structural  biology through the application of lanthanide" -->  " in  structural  biology by applying lanthanide"

"provides the possibility to switch" --> "allows switching"

"we used the transformation of equation (2)" --> "we used equation (2) transformation"

"Matrix‐assisted laser desorption ionization time‐of‐flight (MALDI‐TOF) mass spectra were measured on a Bruker Daltonics Ultraflex spectrometer. " --> "The mass spectra of matrix-assisted laser desorption ionization time‐of‐flight (MALDI‐TOF) were measured on a Bruker Daltonics Ultraflex spectrometer."

" range  of  250‐900  nm" --> "250‐900  nm range"

"at  303  К  with  the  use  of  the  residual solvent" --> "at  303  К  with the  residual solvent"

"Importantly,  the  dipolar  component" --> "Notably,  the  dipolar  component"

Author Response

Response to Reviewer 1 Comments

Comments and Suggestions for Authors

This manuscript "Site-selective solvation-induced conformational switching of heteroleptic heteronuclear Tb(III) and Y(III) trisphthalocyaninates for the control of their magnetic anisotropy" presents an interesting study on the correlation between the structure and magnetic properties of heteroleptic crown-substituted trisphthalocyaninates [C4]M*[C4]M(Pc). The authors demonstrate how changes in the rotational state of the phthalocyanine ligands can be used to tune the axial anisotropy of the Tb ions, resulting in an increase of 25%. This work is well-written and provides a thorough analysis of the results, as well as an insightful discussion on the implications of this work and potential applications in the design of magnetic materials. The authors have done a great job in this work, though there are a few minor revisions that could be made to improve the clarity of the manuscript.

–For example, the "M and M*" should include the symbols for the elements, and the description of the native state of the complex could be more concise.

RESPONSE: The comment is not clear - we give the labels of both metal centers throughout the manuscript when it is necessary to specify the position of the Tb3+ ion with respect to the ligand environment, and the general designation [B4]M[B4]M*[C4] is used as a general label for this type of complex, but it is always followed by the specification of the M and M* metal centers.

As for the native state of the complexes, we are not sure what you mean. If you are asking about the conformation of the complex in the absence of solvent bound to substituents, then we can't give a firm answer to this question as there are no structural data available for this type of complex, either in the gas phase or in the solvent-free solid state. However, we can expect that in this case both [B4]M[B4] and [B4]M*[C4] pairs might be in gauche conformations - this cautious assumption can be ruled out from our previous DFT modelling of the structure and interligand interactions in alkoxy-substituted triple-deckers [10.3390/molecules27196498]. However, this speculative discussion of the hypothetical state of the complexes in the solvent-free state is beyond the scope of this work where the well-defined solvated state is analyzed.

–Additionally, it may be helpful to add a brief description of the NMR measurements to section 4.2. Methods (Pulse width, Number of scans, Spectral width, Relaxation Delay etc).

RESPONSE: We added the required information to Section 4.2:

1H NMR spectra were acquired with standard Bruker zg30 pulse program for 30 degree flip angle. The acquisition and relaxation delays were 1 s and number of scans was 32. The spectra were recorded with 192298 points resolution and line broadening factor -0.5 Hz for Fourier transformation.

1H-1H COSY spectra were a acquired with standard Bruker gradient-enhanced quantum-filtered COSY pulse sequence cosygpqf. The acquisition and relaxation delays were 0.137 s and 1 s in each scans, respectively, with 4 scans per increment. The spectra were recorded with 16384 x 512 points resolution.

–Please add information concerning study of lanthanide complexes in buffer solutions to introdaction [10.1021/acs.inorgchem.0c03082].

RESPONSE: We added the required reference to the manuscript – see Ref. 8

Overall, this is an interesting and well-written manuscript that provides an in-depth look at the correlation between the structure and magnetic properties of heteroleptic crown-substituted trisphthalocyaninates. With the suggested revision, this manuscript is ready for publication.

RESPONSE: We would like to thank the reviewer for the consideration of our work and valuable comments!

Comments on the Quality of English Language

Scientific sound corrections (not essential, but will help make your manuscript clear to understand):

"in  structural  biology through the application of lanthanide" -->  " in  structural  biology by applying lanthanide"

RESPONSE: Corrected

"provides the possibility to switch" --> "allows switching"

RESPONSE: We propose the following correction “In turn it switches the coordination polyhedron of the metal center M from square-antiprismatic (SAP) to distorted prismatic (DP).”

"we used the transformation of equation (2)" --> "we used equation (2) transformation"

RESPONSE: We prefer to keep our version

"Matrix‐assisted laser desorption ionization time‐of‐flight (MALDI‐TOF) mass spectra were measured on a Bruker Daltonics Ultraflex spectrometer. " --> "The mass spectra of matrix-assisted laser desorption ionization time‐of‐flight (MALDI‐TOF) were measured on a Bruker Daltonics Ultraflex spectrometer."

RESPONSE: We prefer to keep our version

" range  of  250‐900  nm" --> "250‐900  nm range"

RESPONSE: Corrected

"at  303  К  with  the  use  of  the  residual solvent" --> "at  303  К  with the  residual solvent"

RESPONSE: Corrected

"Importantly,  the  dipolar  component" --> "Notably,  the  dipolar  component"

RESPONSE: We prefer to keep our version, as it is really important

Reviewer 2 Report

The fundamental and practical significance of the MS (molecules-2397386) is well demonstrated in the Introduction section. This work is the continuation of the extensive work of this scientific group on highlighting structure-property correlations in the heteroleptic heteronuclear lanthanide complexes with phthalocyaninates. However, the novelty of the present work is clearly demonstrated in the Introduction section. Results section is also good written, and both fundamental background and experimental data are enough discussed to make it clear that indeed the solvent induced changes in the conformations are enough to trigger the changing in the axial anisotropy term by 25 and 10%. The discussion of the results is separated in the separate section, the latter section discusses the previous results on inducing the similar changes in axial anisotropy term by coordination of K+ ions via the crown rings. However, readers need more detailed discussion of the obtained changes in correlation with literature data with highligting why changes by 25% are spectacular. It is also worth noting that the parafraph discussing the practical  significance of the quantitative evaluation of axial anisotropy terms is better to transfer into the Introduction section.

Author Response

Response to Reviewer 2 Comments

Comments and Suggestions for Authors

The fundamental and practical significance of the MS (molecules-2397386) is well demonstrated in the Introduction section. This work is the continuation of the extensive work of this scientific group on highlighting structure-property correlations in the heteroleptic heteronuclear lanthanide complexes with phthalocyaninates. However, the novelty of the present work is clearly demonstrated in the Introduction section. Results section is also good written, and both fundamental background and experimental data are enough discussed to make it clear that indeed the solvent induced changes in the conformations are enough to trigger the changing in the axial anisotropy term by 25 and 10%. The discussion of the results is separated in the separate section, the latter section discusses the previous results on inducing the similar changes in axial anisotropy term by coordination of K+ ions via the crown rings. However, readers need more detailed discussion of the obtained changes in correlation with literature data with highligting why changes by 25% are spectacular. It is also worth noting that the parafraph discussing the practical  significance of the quantitative evaluation of axial anisotropy terms is better to transfer into the Introduction section.

RESPONSE: We would like to thank the reviewer for the consideration of our work and valuable comments!

In accordance with your recommendation, we transferred the paragraph concerning the correlation between the anisotropy and temperature sensitivity in NMR to the introduction to emphasize the practical importance of the obtained results from the very beginning of the manuscript.

We also added brief comparison of two examples of single molecule magnets, where the energy barriers for spin reversal and the axial magnetic anisotropies are compared, thus suggesting that the increase of anisotropy might be used as a tool to tune the SMM behavior.

Reviewer 3 Report

This manuscript presents the synthesis and characterization of an interesting series of new compounds of isomeric heteronuclear terbium(III) and yttrium(III) triple‐decker phthalocyaninates [(BuO)8Pc]M[(BuO)8Pc]M*[(15C5)4Pc] (M = Tb, M* = Y or M = Y, M* = Tb, [(BuO)8Pc]2‐ – octa‐n‐butoxyphthalocyaninato‐ligand, [(15C5)4Pc]2‐ – tetra‐15‐ crown‐5‐phthalocyaninato‐ligand). The authors detailed both the synthesis of these compounds and characterized them with a number of experimental methods.  They then explored their properties in a number of solvent environments and found that these complexes exhibited solvation‐induced switching in which the conformers where both metal centers are in square‐antiprismatic structures are stabilized in toluene but in dichloromethane the metal centers M and M* are in distorted prismatic and antiprismatic structures, respectively as determined from an in depth analysis of the lanthanide‐induced shifts in their 1H NMR spectra that allow one to deduce the axial component of the magnetic susceptibility tensor χax, Tb and find that this term is especially sensitive to the conformational switching when the terbium(III) ion is situated in the switchable ʺMʺ site. The new findings in this manuscript will help in better modulating the magnetic properties of lanthanide complexes with phthalocyanine ligands and be of interest to a number of readers of MOLECULES.  All of the experiments and the interpretation of the results appear to have been carefully done and the results seem reasonable.  I recommend publication after minor revision to address the comments/suggestions below:

1.     In Table 1 the authors may want to highlight those entries they consider to be more important to take note of or which are explicitly discussed in the main text of the manuscript. 

2.    For Figure 2 the panels could be made larger to fit the full width of the page and placed above/below each other.  The main absorption bands could also have some description label of their assignment of nature added above their wavelength number already given in the figure.

3.    For Figure 5 (a,b) please indicate the uncertainties of the data points in either the figure or in the figure caption.

just minor editing needed

Author Response

Response to Reviewer 3 Comments

This manuscript presents the synthesis and characterization of an interesting series of new compounds of isomeric heteronuclear terbium(III) and yttrium(III) triple‐decker phthalocyaninates [(BuO)8Pc]M[(BuO)8Pc]M*[(15C5)4Pc] (M = Tb, M* = Y or M = Y, M* = Tb, [(BuO)8Pc]2‐ – octa‐n‐butoxyphthalocyaninato‐ligand, [(15C5)4Pc]2‐ – tetra‐15‐ crown‐5‐phthalocyaninato‐ligand). The authors detailed both the synthesis of these compounds and characterized them with a number of experimental methods.  They then explored their properties in a number of solvent environments and found that these complexes exhibited solvation‐induced switching in which the conformers where both metal centers are in square‐antiprismatic structures are stabilized in toluene but in dichloromethane the metal centers M and M* are in distorted prismatic and antiprismatic structures, respectively as determined from an in depth analysis of the lanthanide‐induced shifts in their 1H NMR spectra that allow one to deduce the axial component of the magnetic susceptibility tensor χax, Tb and find that this term is especially sensitive to the conformational switching when the terbium(III) ion is situated in the switchable ʺMʺ site. The new findings in this manuscript will help in better modulating the magnetic properties of lanthanide complexes with phthalocyanine ligands and be of interest to a number of readers of MOLECULES.  All of the experiments and the interpretation of the results appear to have been carefully done and the results seem reasonable.  I recommend publication after minor revision to address the comments/suggestions below:

RESPONSE: We would like to thank the reviewer for the consideration of our work and valuable comments!

  1. In Table 1 the authors may want to highlight those entries they consider to be more important to take note of or which are explicitly discussed in the main text of the manuscript. 

RESPONSE: We marked selected resonance signals with bold font styles to show the cases where LIS values change their signs.

  1. For Figure 2 the panels could be made larger to fit the full width of the page and placed above/below each other.  The main absorption bands could also have some description label of their assignment of nature added above their wavelength number already given in the figure.

RESPONSE: We modified the image wo that it fits the page width and added the labels N, B and Q to mark the nature of corresponding bands.

  1. For Figure 5 (a,b) please indicate the uncertainties of the data points in either the figure or in the figure caption.

RESPONSE: The slopes of the regression lines in Figures 5a and 5b correspond to 12π·cax, so they have converted to cax and plotted with corresponding uncertainties in Figure 5c. These values are already given in the text (lines 186 and 201 in the revised manuscript), so we see no need to repeat them in Figures 5a,b or their captions.